# Global Impairment of Immediate-Early Genes Expression in Rett Syndrome Models and Patients Linked to Myelination Defects

**DOI:** 10.3390/ijms24021453

**Published:** 2023-01-11

**Authors:** Paolo Petazzi, Olga Caridad Jorge-Torres, Antonio Gomez, Iolanda Scognamiglio, Jordi Serra-Musach, Angelika Merkel, Daniela Grases, Clara Xiol, Mar O’Callaghan, Judith Armstrong, Manel Esteller, Sonia Guil

**Affiliations:** 1Josep Carreras Leukemia Research Institute, School of Medicine, University of Barcelona, Carrer Casanova 143, 400° floor, 08036 Barcelona, Spain; 2RICORS-TERAV, Instituto de Salud Carlos III, 28029 Madrid, Spain; 3Josep Carreras Leukaemia Research Institute (IJC), Badalona, 08916 Barcelona, Spain; 4Biosciences Department, Faculty of Sciences and Technology (FCT), University of Vic-Central University of Catalonia (UVic-UCC), C. de la Laura, 13, 08500 Vic, Spain; 5Cancer Epigenetics and Biology Program (PEBC), Bellvitge Biomedical Research Institute (IDIBELL), L’Hospitalet de Llobregat, 08908 Barcelona, Spain; 6Fundación San Juan de Dios, 08950 Barcelona, Spain; 7Servei de Medicina Genètica i Molecular, Institut de Recerca Pediàtrica, Hospital Sant Joan de Déu, 08950 Barcelona, Spain; 8Clínica Rett, Neurology Department, Hospital Sant Joan de Déu, 08950 Barcelona, Spain; 9CIBER-ER (Biomedical Network Research Center for Rare Diseases), Instituto de Salud Carlos III, 28029 Madrid, Spain; 10Centro de Investigacion Biomedica en Red Cancer (CIBERONC), 28029 Madrid, Spain; 11Institució Catalana de Recerca i Estudis Avançats (ICREA), 08010 Barcelona, Spain; 12Physiological Sciences Department, School of Medicine and Health Sciences, University of Barcelona (UB), 08907 Barcelona, Spain; 13Germans Trias i Pujol Health Science Research Institute, Badalona, 08916 Barcelona, Spain

**Keywords:** Rett syndrome, MeCP2, EGR2, IEGs, myelination

## Abstract

Rett syndrome (RTT) is a severe neurodevelopmental disease caused almost exclusively by mutations to the *MeCP2* gene. This disease may be regarded as a synaptopathy, with impairments affecting synaptic plasticity, inhibitory and excitatory transmission and network excitability. The complete understanding of the mechanisms behind how the transcription factor MeCP2 so profoundly affects the mammalian brain are yet to be determined. What is known, is that MeCP2 involvement in activity-dependent expression programs is a critical link between this protein and proper neuronal activity, which allows the correct maturation of connections in the brain. By using RNA-sequencing analysis, we found several immediate-early genes (IEGs, key mediators of activity-dependent responses) directly bound by MeCP2 at the chromatin level and upregulated in the hippocampus and prefrontal cortex of the *Mecp2*-KO mouse. Quantification of the IEGs response to stimulus both in vivo and in vitro detected an aberrant expression pattern in MeCP2-deficient neurons. Furthermore, altered IEGs levels were found in RTT patient’s peripheral blood and brain regions of post-mortem samples, correlating with impaired expression of downstream myelination-related genes. Altogether, these data indicate that proper IEGs expression is crucial for correct synaptic development and that MeCP2 has a key role in the regulation of IEGs.

## 1. Introduction

Rett syndrome (RTT) is the major cause of mental disability in women after Down syndrome, and is characterized by several cognitive and motor impairments acquired in early childhood [1]. At this stage of brain development, sensory experience is centered around the modulation, maturation and elimination of excitatory synapses, and promotion of the development of inhibitory synapses [2]. Disruption of the normal process of experience-dependent synaptic development may lead to an imbalance between excitatory and inhibitory neurons. This in turn, can result in the cognitive impairment that is seen, in autism spectrum disorder (ASD) and other disorders with overlapping features, including RTT [3]. Consistent with the hypothesis that RTT arises as a disorder of neuronal maturation, the lack of MeCP2 does not seem to alter the proliferation rate and cell density of neural stem cells from the dentate gyrus (DG) of the hippocampus [4]. On the other hand, there is supporting evidence suggesting that impairments in both synapse maturation and function are involved in RTT physiopathology [5]. Firstly, autopsy studies have shown reduced dendritic branching and increased cell-packing density in the brains of RTT patients [6]. This is further supported by characterization of RTT mouse models at the synaptic level. So far, reported defects include: reductions in dendritic spine density [7,8], decrease in glutamatergic synapse number [9], and changes in the strength and/or number of GABAergic inhibitory synapses [10].

Mutations to the *MeCP2* gene account for more than 90% of RTT cases. MeCP2 is an abundant nuclear protein that was initially considered a negative regulator of transcription, due to its high affinity to methylated cytosine (5 mC) [11,12]. However, in the last decade several studies have elegantly deepened the knowledge behind the biological function of MeCP2. As a result, a multi-functional, global regulatory role for the protein has emerged. It is now known, MeCP2 is able to modulate the expression (activating or repressing) of a wide range of genes through interaction with several partners [13], and through post-translational modifications at several residues [14]. Besides the high affinity to 5 mC, MeCP2 has been demonstrated to be the major 5-hydroxymethylcytosine (5 hmC)-binding protein in the brain (an epigenetic mark enriched in active genes), with a specific selectivity towards CA repeats [15,16]. In addition to this, MeCP2 can bind to DNA with low specificity through its highly conserved AT-hook domains [17]. Impairment of one of the AT-hook domains within MeCP2 appears to impair its DNA binding and chromatin compaction capabilities resulting in the loss of ATRX protein localization at pericentric heterochromatin. Besides these genomic substrates, the function of MeCP2 may be modified, depending on its bound protein partner. MeCP2 interactors encompass the transcriptional activator CREB [18], the NCoR/SMRT repressor complex [19], the histone deacetylase HDAC1/2 [20], the regulator of alternative splicing YB1 [21] and the DNA methyltransferase DNMT1 [22], among others.

With the aim of gaining new insights on the tissue-specific roles of MeCP2 in the brain, here we have focused on three distinct brain regions: the hippocampus, the prefrontal cortex and the cerebellum. RNA-seq analysis revealed that many immediate-early genes (IEGs), which are crucial players in neuronal plasticity and cognitive function [23], are mis-regulated in the brain of a RTT mouse model, human post-mortem samples and the peripheral blood of RTT patients. In addition, the lack of MeCP2 results in an impaired response to stimuli of IEG expression in vitro and in vivo. Furthermore, dysfunction of IEGs impacts on some transcriptional targets, including genes involved in the myelination process. To summarize, our results contribute to the understanding of the critical involvement of MeCP2 in synaptic plasticity, which in turn is key to higher cognitive abilities, learning and memory skills.

## 2. Results

### 2.1. RNA-Seq Analysis Reveals Dysregulation of IEGs in a Mouse Model of RTT

We focused our attention on RNA-seq data from the hippocampus (HIP) and frontal cortex (FC) of symptomatic, 8-week-old *Mecp2*-KO mice, and compared gene expression levels to those of wild type littermates of the same age (Appendix A). Of note, examination of upregulated genes in both regions revealed an enrichment in pathways belonging to NGF-stimulated transcription and NTRK1 signaling (Appendix A–d), which are cellular programs involved in neuronal differentiation and neurite outgrowth, often driven by immediate-early genes (IEGs) [2]. For example, all significantly enriched reactome categories in the hippocampus were contributed by the following IEGs: *Egr1*, *Egr2*, *Arc*, *Fosb* and *Fos* (Appendix A). In addition to these, common upregulated genes in the hippocampus and frontal cortex of *Mecp2*-KO mice included *Nr4a1*, *Npas4* and *Junb* (Appendix A, Table 1). IEGs are rapidly and transiently induced by sensorial, emotional or cognitive stimuli. This activity-dependent family of genes encodes several functionally different products such as secreted proteins (cytokines), cytoplasmic enzymes, and ligand-dependent or inducible transcription factors. IEGs act as signal mediators between the first and second messengers, usually extracellular factors (neurotransmitters) and cytoplasmic protein kinases, respectively, and the late neuronal downstream targets that directly regulate the extent of neuronal plasticity changes and memory function (reviewed in [23]). In concordance with our RNA-seq results, activity-dependent genes have been shown before to be aberrantly expressed in a mouse model of Rett syndrome [24]. In addition, the examples of cellular receptors whose inputs crosstalk with the expression of IEGs include AMPA receptors [25,26], which we have recently shown to be dysregulated in human cellular models and post-mortem samples of Rett syndrome [27]. Thus, we reasoned that this class of genes could be of prime interest for the pathophysiology of RTT.

Among the upregulated IEGs in the *Mecp2*-KO brains we found *Fos*, *Fosb* and *Junb* (members of the AP-1 complex, a known transcription factor that regulates gene expression in response to a multitude of stimuli, including growth factors and neurotransmitters), and *Egr1* and *Egr2* (two members of the Early growth response gene group, another class of transcriptional regulators involved in the neuronal response stimuli [28]). In addition, the Nr4a family members *Nr4a1* and *Nr4a3* are also present. This group of genes has been implicated in hippocampus-dependent memory formation and consolidation [29]. Finally, we also detected upregulation of *Arc*, which encodes a post-synaptic protein involved in synaptic plasticity [23], and *Npas4*, a transcription factor that promotes the formation and maintenance of inhibitory synapses [30] (Table 1).

In order to confirm our results, we assessed the expression of the differentially-expressed candidates by RT-qPCR in independent samples. Consistent with the findings from the RNA-sequencing analysis, most IEGs showed significant upregulation in the hippocampus and frontal cortex (Figure 1a,b). By contrast, no change was observed in the transcript levels of *Arc* in either of the two brain regions (Figure 1a,b). We further confirmed by Western Blot that the levels of some key IEGs were upregulated at the protein level (Figure 1c). In the case of *Egr2*, we could also clearly detect increased EGR2 protein expression by immunostaining of hippocampal slices from *Mecp2*-KO mice (Figure 1d and Appendix A). Even though we could not detect a significant increase in the levels of NPAS4 protein, neither by Western blot nor by immunostaining (Figure 1c and Appendix A), exploration of other genes related to synaptic development detected alterations in the expression of CCAAT enhancer binding proteins alpha and beta (Cebpa and Cebpb) (Appendix A), suggesting global dysregulation of learning-induced IEGs.

### 2.2. MeCP2 Has a Potential Repressive Role on IEGs

Since we detected impairment in the expression of several members of the IEGs family in the *Mecp2*-KO brain, we decided to interrogate the potential of direct regulation by MeCP2 on these genes. Albeit multifunctional, in several studies MeCP2 action has been linked to transcriptional repression. This spans across its discovery as a methyl-CpG binding protein [11], to the finding that MeCP2 is able to interact with HDAC2, Sin3b [12] and NcoR/Smrt [19] co-repressor complexes. Apart from this, MeCP2 regulation has also previously been correlated with changes in IEGs expression in neurons [31,32], although its exact molecular underpinnings are unclear.

To determine the involvement of MeCP2 in the regulation of IEGs we analyzed their transcription start site (TSS) regions by MeCP2 chromatin immunoprecipitation (ChIP) both from the prefrontal cortex and hippocampus of wild-type mice. Five regions along each IEG TSS were selected for ChIP-qPCR analysis. Consistent with previous studies on MeCP2 occupancy in neurons [33], MeCP2 binding was observed throughout all the promoter regions, while IgG controls showed almost undetectable ChIP-qPCR products in both FC and HIP (Figure 2a–h). This is consistent with the findings from publicly available ChIP-seq datasets (Appendix A) [34]. Noteworthy, in every IEG analyzed by ChIP, the level of MeCP2 occupancy on the TSS and proximal regions was higher in HIP than FC, although they share the same binding pattern along the 4 Kb genomic windows (Figure 2a–h). Furthermore, in *Fos*, *Junb*, *Nr4a1*, *Npas4*, *Fosb* and *Egr1*, both FC and HIP show a reduction in MeCP2 binding to the regions associated with high CpG content (Figure 2a, b, d–f and h). The fact that MeCP2 is enriched within low CpG-content regions is likely due to the methylated status of CpG dinucleotides outside CpG islands (CGI, which are expected to be unmethylated in the case of these actively transcribed genes). To further prove that MeCP2 may be crucial for the proper regulation of these genes, we investigated the changes in the chromatin structure around the IEGs promoter upon MeCP2 loss. With this aim, we performed a chromatin accessibility assay by means of micrococcal nuclease (MNase) digestion of nuclei, prepared from the HIP of WT and *Mecp2*-KO animals. After digestion with MNases, the chromatin fragments were subjected to qPCR. The accessibility of *Fos*, *Junb*, and *Npas4* promoters was significantly increased in *Mecp2*-KO hippocampi (Figure 2i), indicating a more open chromatin state in the absence of MeCP2. Taken together, these results demonstrate that MeCP2 (i) binds to IEGs promoters, with a preference for regions outside the CGI islands, and (ii) its loss enhances chromatin accessibility, suggesting that MeCP2 plays a repressive role in the expression of IEGs in the hippocampus.

### 2.3. IEGs Response to Activation Is Misregulated in Mecp2-KO Neurons

Given the fact that IEGs are activity-dependent genes, and their expression is greatly enhanced after applying a stimulus (i.e., synaptic transmission in neurons), we next tested the transcriptional impairment caused by the loss of MeCP2 upon the trigger of an activating stimulus. With this aim, we cultured primary neurons derived from newborn WT and *Mecp2*-KO mice. Since we sought for a homogeneous neuronal model, we performed cortical and hippocampal primary neuron cultures to resemble the neuronal population of the respective brain areas. As expected, both cortical and hippocampal *Mecp2*-KO neurons showed no expression of MeCP2 (Figure 3a). Next, we decided to treat our cultures with forskolin, a well-known adenylate cyclase activator that induces cAMP levels, which in turn triggers the activation of the PKA pathway [35]. Forskolin is widely used to elicit chemical long-term potentiation (cLTP) [36] and modulate the synaptic efficacy of excitatory glutamatergic synapses in the mammalian brain [37]. Therefore, to investigate IEGs expression in our cultures we performed the analysis of expression over a time-course following forskolin treatment. As shown in Figure 3b, 1 h after treatment with the drug, the medium was replaced with fresh media (without forskolin) and the recovery to the basal level was assessed. *Fos*, *Junb*, *Egr2* and *Npas4* displayed differential response patterns after forskolin treatment and withdrawal among different tissues and experimental conditions. In general, *Fos*, *Junb*, *Egr2* and *Npas4* expression levels in the *Mecp2*-KO hippocampal neurons were higher than those of wild-type (Figure 3c). Of note, one hour after forskolin withdrawal, the expression of these genes continued to increase in *Mecp2*-KO neurons, while in the wild-type they did not change or even decrease. As a result, the expression levels of the four IEGs in wild-type and *Mecp2*-KO hippocampal neurons one hour after forskolin withdrawal are significantly different, suggesting that MeCP2 may be critical for the recovery of basal levels of IEGs expression after a stimulus, at least in the hippocampus, thereby contributing to the reduced synaptic plasticity in RTT. Intriguingly, the situation is the opposite in cortical neurons, where *Fos*, *Junb*, *Egr2* and *Npas4* are less expressed after forskolin withdrawal in *Mecp2*-KO samples (Figure 3d).

In summary, our results show that MeCP2 is fundamental for the proper recovery of IEGs basal expression levels in hippocampal neurons stimulated with forskolin. However, the same genes display decreased responsiveness in cortical neurons, suggesting that the MeCP2-mediated regulation of the kinetic of IEGs expression is context-dependent.

### 2.4. IEGs Are Over-Activated in Kainate-Stimulated Hippocampus In Vivo

IEGs are known to be induced after stress, penetrating brain injury, systemic phencyclidine administration and cerebral ischemia, amongst other conditions [38,39,40]). Kainic acid (KA) is a potent epileptogenic drug that induces excitatory neurons by activating kainate glutamate receptors and, to a minor extent, AMPA channels [41]. KA-induced seizures produce damage in specific brain regions, with hippocampal regions being most severely affected [42]. Several studies have shown that the main target of KA, when administered in vivo, is the CA3 region of the hippocampus (where high levels of kainate receptors are found [41]), resulting in activation of a myriad of genes, including IEGs [43]. Thus, to further prove the impairment in IEGs expression in the brain of *Mecp2*-KO mice, we treated WT and MeCP2-deficient animals with either KA or saline solution, and measured IEGs expression in the hippocampus and frontal cortex. While no changes were detected in *Mecp2* mRNA levels between KA- and saline-injected samples (Figure 4a,c), we observed a significant increase in *Junb* and *Egr2* expression in the hippocampus of *Mecp2*-KO mice treated with KA, when compared with treated WT animals and saline controls (Figure 4b). By contrast, in this same region, *Fos* and *Npas4* were not induced at higher levels in *Mecp2*-KO compared with WT animals (Figure 4b). None of the analyzed IEGs were further enhanced in the frontal cortex of *Mecp2*-KO brains when compared to WT animals (Figure 4d). Altogether, these observations confirm the data from primary neuronal cultures and reinforce the finding that activity-dependent induction of *Junb* and *Egr2* are impaired in the hippocampus of a RTT mouse model.

### 2.5. IEGs Are Dysregulated in Different Regions from Human RTT Post-Mortem Brains and Patient’s Peripheral Blood

Having seen dysregulation of IEGs in a mouse model of Rett syndrome, we next analyzed their altered expression in human post-mortem samples, which represent the closest reality to the end-point of the disorder. We assessed IEGs expression in the hippocampus and cerebellum from RTT patients and normal, age-matched brain regions (aged 15–22 years). At the transcript level, we detected higher levels in RTT brains for all IEGs tested (*EGR2, JUN, FOS, JUNB*), with *EGR2* and *FOS* significantly altered in the hippocampus (Figure 5a) and *JUN* and *JUNB* markedly increased in the cerebellum (Figure 5c). Corresponding protein levels could also be analyzed in the same samples, and EGR2 and JUN displayed the highest upregulation in both tissues (Figure 5b,d). Similar upregulation at the protein level was observed in RTT frontal cortex (Appendix A). By contrast, NPAS4 was consistently downregulated in all post-mortem brain regions analyzed (Appendix A), indicating differential dysregulation for individual IEGs in patients, possibly linked to their specific roles in cognitive-related processes. To further prove the impact of MeCP2 regulation on these genes in human models, we used neural progenitor cells depleted of MECP2 by means of CRISPR/Cas9 [27]. The re-expression of *MECP2* in these cells markedly downregulated the levels of *FOS*, *JUNB* and *EGR2* mRNAs (Figure 5e), confirming its regulatory role over IEGs in the human context. Furthermore, IEGs alterations can be observed in the peripheral blood of RTT patients too: in this case, *EGR2* mRNA is upregulated and both *JUN* and *JUNB* mRNAs are downregulated, suggesting a global dysregulation within this family of genes and their potential value as biomarkers (Figure 5f).

Finally, EGR2 dysfunction has been linked to immune dysregulation and altered pro-inflammatory cytokines levels [44], a feature also described in *Mecp2*-KO mouse models and patients [8,45,46,47]. We confirmed the increased expression of some pro-inflammatory cytokines in the post-mortem RTT brains (Appendix A), as previously reported in RTT [8], which, we predict with future research, this potential molecular link between EGR2 function, neuroinflammation and microglial infiltration will be resolved.

### 2.6. Aberrant EGR2 Expression Correlates with Altered Myelination Pathways in RTT Patients

In order to get further insights into the contribution of IEGs hyperactivation on the RTT phenotype and taking into account, (i) such as in other neurological disorders, RTT displays glial abnormalities, with myelination being altered in *Mecp2*-KO mice [48,49] and (ii) EGR2 is highly expressed in Schwann cells and oligodendrocytes and is a master regulatory gene for myelination through partnering with NAB proteins [50,51] (Figure 6a), we next explored the expression of key markers of the myelin production pathway in the mouse and human RTT samples (Figure 6b,d). In the mouse *Mecp2*-KO model, the myelin basic protein *MBP* and the myelin proteolipid protein *PLP1* genes (two major constituents of the myelin sheath of oligodendrocytes and Schwann cells) displayed little alterations in the hippocampus at the mRNA and protein level (Figure 6b,c). However, there was a clear decrease at the protein level in the 8-week cerebellum (despite no significant changes in the mRNA) (Figure 6e). Similarly, no major changes in *MBP* or *PLP1* mRNA expression were observed in the cerebellum of post-mortem samples (Figure 6f), but both proteins were detected at lower levels in the cerebellum and hippocampus of patients (Figure 6g,h), suggesting that a complex post-transcriptional regulation is involved in the impairment of myelination processes. Remarkably, expression of the two paralogs *NAB1* and *NAB2* (co-transcriptional regulators that inhibit EGR2 function) are detected at lower levels in the post-mortem cerebellum (Figure 6f), confirming impairment of the NAB1/2/EGR2 axis in RTT patients and suggesting the molecular basis for aberrant myelination. Altogether, our data confirm IEGs dysregulation in mouse and human samples as a result of MeCP2 loss-of-function, and reveal abnormal patterns of expression in key genes of the myelin pathway that are under direct control of EGR2 function. This may have an important impact on central and peripheral nervous system development, connectivity and maturation.

## 3. Discussion

Several studies have focused on the identification of key MeCP2 targets in an attempt to explain RTT pathophysiology. In addition to this, meta-analyses of historic transcriptomic and proteomic studies have been performed to elute consistent alterations across different experimental models [52,53]. However, the highly heterogeneous systems used pose great difficulties to draw robust conclusions, from specific cell types. In addition, details about the molecular pathways that contribute the most to the disease process are still unclear. Our transcriptomic approach has focused specifically on a subset of upregulated genes, the immediate-early genes (IEGs) family, which are present in two brain regions highly affected in RTT, the hippocampus and the prefrontal cortex.

There are several papers reporting mis-regulation of one or more members of the IEGs family in RTT [31,54,55,56,57], but the different biological models used and the heterogeneous experimental conditions applied make it difficult to properly review the topic. Here, we provide evidence of a global impairment of IEGs expression in the *Mecp2*-KO mouse. Validation of sequencing data by qRT-PCR confirmed the aberrant expression of *Egr2*, *Nr4a1*, *Nr4a3* and *Egr1* in the HIP and *Fos*, *Junb*, *Npas4* and *Fosb* in both HIP and FC of *Mecp2*-KO mouse. Furthermore, we have been able to detect high levels of expression of IEGs family members in patient’s post-mortem tissue, which altogether indicates robust dysregulation of these important regulatory genes in the disease.

Consistent with the critical finding that MeCP2 binds throughout the whole genome tracking methylated cytosines [33], we confirm the binding of MeCP2 on the IEGs promoters with a de-enrichment in occupancy near the high-CpG-content regions, which is very likely a consequence of unmethylated CpG islands on these highly transcribed genes. Moreover, we have shown that the MeCP2 occupancy pattern along a 4kb-region spanning the TSS is similar between FC and HIP, although the level of MeCP2 binding was higher in the HIP in every IEG promoter analyzed. However, this last observation does not take into account intrinsic differences in tissue composition, for instance the neuron/glia ratio. For this reason, we cannot state that the level of MeCP2 binding on IEGs promoters correlates with the degree of repression of the same genes. Nevertheless, we propose a role for MeCP2 in the regulation of IEGs in light of its binding upon the regulatory regions of this activity-dependent class of genes. The finding that the HIP chromatin was more accessible to MNase digestion in the *Mecp2*-KO brain suggests that decompaction of chromatin is accompanied by IEGs upregulation and further supports our hypothesis, at least in this specific brain region.

IEGs are activity-regulated genes that respond to a myriad of stimuli. In the case of neuronal IEGs the stimulus is represented by synaptic activity. Therefore, to better understand the nature of IEGs impairment in our RTT model, neuronal activity needs to be synchronized by external means. One of the key findings of our study is that hippocampal and cortical neurons differ in the response to forskolin, an adenylate cyclase activator. Four IEGs (*Fos*, *Junb*, *Egr2*, *Npas4*) displayed altered expression in *Mecp2*-KO cultured neurons, and exhibited an aberrant kinetic of recovery to the basal state. It is noteworthy that IEGs activation constitutes a key step for long-term alterations that are necessary for plasticity and memory formation [58]. Defects in the normal activation patterns of IEGs lead to impairment in sensory- and experience-dependent learning mechanisms [23]. In the case of RTT, it is known that overall neuronal activity is decreased in MeCP2 mutant cells. Therefore, our findings from the hippocampus may seem paradoxical, since MeCP2-deficient neurons display increased expression of IEGs. Our data may contribute to explain apparent discrepancies in previous work [59], where an ambiguous state in *Mecp2*-null hippocampal slice preparation was identified [59]. While the intrinsic network activity was hyper-excitable, the local spontaneous post-synaptic excitatory activity was diminished compared to that of wild type. It seems plausible that this hyper-excitation of hippocampal neurons may be due to the increased responsiveness of IEGs. On the other hand, the divergence between in vivo and in vitro data in the case of cortical neurons can partially be explained with the fact that the prefrontal cortex is a concrete area of the adult brain, whereas cortical neurons are derived from the whole neonatal cortex. However, consistent with our in vitro results, Kron et al. [54] reported the reduced expression of *Fos* in forebrain cortices of *Mecp2*-null mice.

The transcription factor EGR2 has been associated with transcriptional activation in cortical cognitive functions associated with attention [60]. This is particularly interesting because these cognitive features are often impaired in autism-spectrum disorders. Moreover, NPAS4 has been potentially implicated with schizophrenia and autism [61], due to its critical involvement in maintaining a balance between excitation and inhibition and thereby contributing to the formation of inhibitory synapses [30]. FOS and JUNB are members of a large family of dimeric protein complexes called AP-1 (activating protein 1). The main scenarios in which AP-1 members seem to play a role are memory formation and behavioral changes related to drug exposure [28]. Intriguingly, the composition of the dimer seems to be the major factor in the regulation of AP-1 in the nervous system. Several studies show that JUNB is a poor transactivator and may act to dampen the response to the more potent transactivators of the family [62].

With the purpose of investigating more in depth IEGs induction impairment in MeCP2-deficient mice, we then analyzed *Fos*, *Junb*, *Egr2* and *Npas4* induction in vivo. Kainic acid- induced seizures provide a valuable tool to this aim, since the administration of this kainate receptor analog provokes extreme neuronal excitation, especially in the HIP, which in turn triggers the activation of several genes, mostly IEGs [63]. We showed that Junb response triggered by kainic acid in *Mecp2*-KO hippocampi was higher than that of control HIP. This represents a further confirmation of the putative role of IEGs expression defects in RTT physiopathology. To gain more insights on this relevant topic, more efforts need to be put in the understanding of the IEGs contribution to RTT, with a close look on JUNB function. Finally, although the pathways activated by forskolin and kainic acid are partially overlapping, the fact that we observed no change in *Fos*, *Egr2* and *Npas4* expression in KA-induced wild type and KO HIP demonstrates that the activation of each IEGs is strictly pathway- and context-dependent.

When analyzing patient’s samples, we have confirmed an increase in IEGs expression in the post-mortem brain tissue and, in the case of *EGR2* mRNA, also in the peripheral blood of patients. These findings warrant future work to analyze the altered levels of circulating IEGs and their potential association with disease outcome, with the aim of establishing robust biomarkers of prognosis, something that is currently lacking in RTT. Of note, EGR2 has been shown to regulate the transcription of key myelin-related genes and to be essential for myelination of the central and peripheral nervous system, and imbalance in EGR2 activity seems to be underlying the abnormal levels of myelination marker genes that we detected in both the animal model and patients. This is concomitant with a decreased expression of the *EGR2* negative regulators *NAB1*/*2* in post-mortem samples, which confirms impairment of the NAB1/EGR2 axis and suggests that NAB1/2 might be targets of MeCP2 dysfunction. Previous studies have reported heterogeneous results regarding the regulation of MeCP2 on key myelin-related genes, possibly derived from the type of sample being analyzed (e.g., cultured oligodendrocytes versus whole brain) [48,64], but at the tissue level a clear reduction on the diameter of myelinated fibers on peripheral nerves of *Mecp2*^+/−^ mice has been shown [65]. The increasing evidence pointing to the importance of glial abnormalities in RTT physiopathology (reviewed in [66]) warrants further investigation on the molecular alterations in myelination pathways downstream of MeCP2, and its specific link to IEGs control.

In summary, this study presents evidence of a misregulation of immediate early genes in the nervous system of a *Mecp2* mouse model, together with confirmation of their altered expression in patients’ samples. Our results may contribute to shed new light on the mechanisms by which synaptic plasticity is impaired in Rett syndrome.

## 4. Materials and Methods

### 4.1. Animals

Experiments were performed on the B6.129P2(c)-MeCP2tm1 + 1Bird mouse model for RTT [67]. The mice were purchased from Jackson Laboratories (stock number 003890) and maintained on a C57BL/6J background. Mice were kept under specific pathogen-free conditions in accordance with the recommendations of the Federation of European Laboratory Animal Science Associations. Lighting conditions (lights on from 08:00–20:00 h) and temperature (22 °C) were kept constant. Animals were allowed ad libitum access to food and water and were inspected every day. Tissue samples were obtained from hemizygous *Mecp2*-null males (*Mecp2*^−/y^, KO) and their wild-type (WT) littermates after establishing RTT-like symptoms in the defective animals (at about 7–8 week of age). All procedures and experiments were approved by the Ethics Committee for Animal Experiments of the IDIBELL Centre (DAAM 7214), under the guidelines of Spanish animal welfare laws. Mice were euthanized in accordance with the Guidelines for Humane Endpoints for Animals Used in Biomedical Research. Tissues were frozen on dry ice immediately after dissection and stored at −80 °C until use.

### 4.2. Human Tissues

Post-mortem brain tissue from control and RTT patients was obtained from the National Institutes of Health (NIH) NeuroBioBank at the University of Maryland (Baltimore, MD, USA) and the Human Brain and Spinal Fluid Resource Center, VA West Los Angeles Healthcare Care Center (Los Angeles, CA, USA), which is sponsored by the National Institute of Neurological Disorders (NMSS), and Stroke (NINDS)/National Institute of Mental Health (NIMH), the National Multiple Sclerosis Society, and the Department of Veterans Affairs. Control samples were from individuals aged 24, 10 and 7 years old. RTT samples were from individuals aged 20, 8 and 6 years old.

### 4.3. Total RNA Isolation

To extract RNA from frozen tissues, mouse prefrontal cortices and hippocampi were ground into powder with mortar and pestle and resuspended in Trizol reagent (ThermoFisher Scientific, Waltham, MA, USA). In the case of primary cultures, Trizol reagent was added directly to the wells and the total RNA purification was performed on the RNA-containing aqueous phase with RNeasy mini kit (Qiagen, Germantown, MD, USA). For RTT post-mortem human brain, the total RNA was extracted with the Maxwell RSC miRNA Tissue Kit (AS1460, Promega, Madison, WI, USA), eluted with RNase-free water and treated with turbo DNase (Ambion). RNA from peripheral blood of Rett patients and controls were extracted with QIAmp RNA blood Mini Kit (Qiagen), following the manufacturer’s protocol. Quantitation and quality check for all samples were performed with Nanodrop and Agilent 2100 Bioanalyzer (Agilent Technologies, Santa Clara, CA, USA). For sequencing purposes, we used RNA samples with RNA integrity number (RIN) above 9.

### 4.4. RNA-Seq Analysis

RNAs from 5 male adult wild-type or Mecp2-KO prefrontal cortices or hippocampi were pooled, and the sequencing library of each RNA sample was prepared with the TrueSeq RNA Sample Preparation kit (Illumina, Cambridge, UK) according to the protocol provided by the manufacturer. Each sample was run in duplicate and subjected to 50 cycles of sequencing in an Illumina Hiseq2000 Sequencer. The RNA-seq data obtained are freely available at the Gene Expression Omnibus (GEO) database: http://www.ncbi.nlm.nih.gov/geo/query/acc.cgi?token=qtsbyccstlgrvut&acc=GSE60219, accessed on 30 September 2018. The RNA-seq reads for all samples were trimmed for adaptors, masked for low-complexity and low-quality sequences and subsequently quantified for transcript expression using Kallisto v0.43.16 [68] and mouse genome version GRCm38 (Ensembl release 102). Gene-level quantification was carried out using the *tximport* Bioconductor package [69] using GRCm38 Ensembl v102 annotations. Differential expression was analyzed using DESeq2 v2.13 [70]. Gene enrichment analyses from significantly altered genes were conducted using Enrichr software (https://maayanlab.cloud/Enrichr/, accessed on 2 November 2022) [71,72].

### 4.5. Quantitative Real-Time PCR

cDNA synthesis was performed with 200–2000 ng total RNA and random hexamers primers using ThermoScript RT-PCR System (Invitrogen, Carlsbad, CA, USA). cDNA (50 ng) was amplified using the SYBR® Green PCR Master Mix (Applied Biosystems, Foster City, CA, USA) in a final volume of 10 µL. Real-time PCR reactions were performed in triplicate on an Applied Biosystems 7900HT Fast Real-Time PCR system. All primer pairs were designed with Primer 3 software and validated by gel electrophoresis to amplify specific single products. PCR cycles were divided into initial denaturation at 95 °C for 5 min, followed by 40 cycles of 95 °C, for 30 s, 60 °C, for 30 s and 72 °C for 30 s. All data were acquired and analyzed with QuantStudio Design & Analysis Software v.1.3.1 and normalized with respect to *Rps28* and *Rpl38* as endogenous controls in mouse samples and *L13* and *PPIA* in human samples. Relative RNA levels were calculated using the comparative Ct method (DDCt). All primers sequences are available upon request.

### 4.6. Chromatin Immunoprecipitation

The procedure was conducted as described before [73]. The brain tissues from 8-week mice were cross-linked with 1% formaldehyde for 8 min and the reaction was blocked by adding glycine to a final concentration of 0.125 M. After washing two times with ice-cold PBS, cell pellets were resuspended in cell lysis buffer (HEPES 5 mM, KCl 85 mM, NP40 0.5% pH 8.0) supplemented with protease inhibitor cocktail (Complete EDTA-free, Roche Diagnostics GmbH, Mannheim, Germany) and the lysate was homogenized with a douncer to facilitate cell membrane break. The nuclear pellet was then resuspended in Nuclei lysis buffer (TRIS-HCl 50 mM, EDTA 10 mM, SDS 1% pH 8.1) and subsequently sonicated with Bioruptor (Diagenode, Seraing, Belgium) for 30 min (30 s ON, 30 s OFF cycles). The chromatin size of the fragments obtained was 150–500 bp. Samples were diluted with Dilution buffer (SDS 0.01%, Triton X-100 1.1%, EDTA 1.2 mM, NaCl 165 mM, TRIS-HCl 16.7 mM pH 8.1). Magnetic beads were used for the pre-clearing of diluted chromatin (over- night at 4 °C) and anti-MeCP2 (m9317, Sigma-Aldrich, St Louis, MO, USA). Non-related mouse IgG antibody (12–371, Merck Millipore, Burlington, MA, USA) was used as a negative control. The Beads-Antibody complexes were then incubated with pre-cleared chromatin for 2 h at 4 °C in agitation. The immune-complexes were washed: twice with low salt Buffer (TRIS-HCl 50 mM pH 8.0, NaCl 150 mM, SDS 0.1%, NP-40 1%, EDTA 1 mM, Deoxicolate Na 0.5%), twice with high Salt Buffer (TRIS-HCl 50 mM pH 8.0, NaCl 500 mM, SDS 0.1%, NP-40 1%, EDTA 1 mM, Deoxicolate Na 0.5%), twice with LiCl Buffer (TRIS-HCl 50 mM pH 8.0, LiCl 250 mM, SDS 0.1%, NP-40 1%, EDTA 1 mM, Deoxicolate Na 0.5%) and twice with TE Buffer (TRIS-HCl 10 mM pH 8.0, EDTA 0.25 mM). Cross-linked chromatin was then eluted from the magnetic beads by adding elution Buffer (NaHCO3 100 mM, SDS 1%). Samples were de-crosslinked overnight at 65 °C and incubated with Proteinase K at 50 ug/mL final concentration for 1 h. Finally, DNA was purified with PCR purification kit (Qiagen).

Analysis of MeCP2 enrichment on gene promoters from public ChIP-seq datasets were obtained from the Short Read Archive (SRA) (accession GSE139509 [34]), and processed with nf-core/chip-seq pipeline v.2.0.0 (https://nf-co.re/chipseq, accessed on 20 October 2022). Briefly, Fastq files were trimmed using Trimgalore and aligned with BWA against mouse genome assembly mm10 using default settings. Genome coverage (bigWig) was computed with bedTools for each replicate. To increase signal strength, we merged replicates and re-computed read coverage using deepTools utility ‘bamCoverage’ v.3.3.1 (parameters: --binSize 25 -normalizeUsing BPM -scaleFactorsMethod None -smoothLength 60 -extendReads 250 -centerReads). Gene-specific read density profiles were produced with deepTools utilities ‘computeMatrix’ (reference-point) and ‘plotProfile’ using mouse genome annotation Gencode vM23.

### 4.7. Chromatin Accessibility Assay

Prefrontal cortices and hippocampi from 8-week mice were homogenized on ice using a douncer in 20 volumes of buffer A (0.25 M sucrose, 5 mM MgCl2, KCl 25 mM, Tris-HCl 20 mM ph 7.5, 0.1% Triton) and then incubated on ice during 5 min. After that, samples were centrifuged at 1000× *g* during 10 min at 4 °C. Pellets were resuspended with buffer A (without Triton) and Optiprep (Sigma) was added. Nuclei were then centrifuged at 3200× *g* for 20 min at 4 °C and the resulting pellets were finally resuspended in 1 volume buffer B (50 mM NaCl, 10 mM PIPES pH 6.8, 5 mM MgCl2, 1 mM CaCl_2_) and pre-warmed at 37 °C for 5 min. Digestions were performed using 2 units of MNase (Roche) per mg of tissue during 15 min. Reactions were blocked on ice with 5 mM EDTA. An aliquot of digested nuclei was taken and used as input. To obtain the chromatin fractions, nuclei were centrifuged at 8000× *g* for 10 min, supernatant corresponds to S1 phase. The digested DNAs were then mixed with Laemmli buffer and warmed at 65 °C. After that, proteins were precipitated at 4 °C with 120 mM KCl and the DNA-containing supernatant was used for Real-time PCR. 0.5 ng of DNA were used for each PCR. Qubit 2.0 Fluorometer and dsDNA Broad Range Assay reagents (Invitrogen, Carlsbad, CA, USA) were used to quantify DNA.

### 4.8. Primary Neuronal Hippocampal and Cortical Cultures

Dissociated cortical and hippocampal neurons were prepared from newborn mice as previously described [74]. Cultures were maintained in Neurobasal A medium supplemented with B27, antibiotic-antimycotic and Glutamax™ (ThermoFisher Scientific). Cytosine arabinoside (Sigma) was added to a 5 μm final concentration at 1 DIV to inhibit the proliferation of dividing non-neuronal cells. Neurons were plated at 100,000–200,000 per well in a 12-well plate. Half medium was replaced with an equal volume of fresh warm medium at 4 DIV. For neuronal stimulation, cultures were incubated with 50 μm forskolin (Sigma) at 11 DIV for 1 h and then the media was replaced with fresh one.

### 4.9. In Vivo Kainic Acid Administration

In vivo stimulation was induced by intraperitoneal administration of symptomatic mice 7–8 weeks of age with kainic acid (KA) (15 mg/kg) in PBS. Control animals received saline only. The efficiency of the treatment was assessed observing seizures in KA-treated animals. Mice were sacrificed 1 h after injection and the brains were rapidly dissected. Tissues were frozen on dry ice immediately after dissection and stored at −80 °C until use.

### 4.10. Western Blot

Mouse tissue FC or HIP were resuspended in Laemmli buffer (2% SDS, 10% glycerol, 60 mM Tris-Cl [pH 6.8], and 0.01% bromophenol blue), sonicated, and boiled for 10 min at 70 °C.

Brain post-mortem tissue was ground into powder with mortar and pestle and resuspended in CelLytic^TM^ MT Mammalian Tissue Lysis/Extraction Reagent (Sigma-Aldrich), with Protease Inhibitor Cocktail (Roche Diagnostics). Pierce BCA Protein Assay Kit (ThermoFisher Scientific) was used for protein quantification. Specific primary antibodies used were Rb-Egr2 (EPR4004, AbCam, Cambridge, UK); Rb-JunB (ab128878, AbCam); Rb-Fos (#2250, Cell Signaling, Danvers, MA, USA); Rb-Jun (#9165S, Cell Signalling); b-actin peroxidase conjugated as control loading (A3854; Sigma-Aldrich). The secondary antibodies used were conjugated to horseradish peroxidase anti-mouse IgG (Sigma-Aldrich; #M7023) and anti-rabbit IgG (Sigma-Aldrich; #A0545). All antibodies were used at dilutions recommended by the manufacturers. In all cases, three post-mortem brain samples for each group were used (Controls and Rett patients). Immobilon Western Chemiluminescent HRP Substrate (#WBKLS0500, Merck Millipore) was used to visualize the proteins. iBright 1500 Imaging System^®^ (ThermoFisher Scientific) was used to obtain the image of the Western Blot and iBright Analysis Software^®^ (ThermoFisher Scientific) was used to determine the density of the bands.

### 4.11. Brain Tissue Immunofluorescence

Tissue immunostaining was carried out as previously described [75]. The following primary antibodies were used: Rb-Egr2 (EPR4004, AbCam), and Rb anti-NPAS4 (orb256724, Biorbyt, Cambridge, UK). Specific secondary antibodies for each condition were used (A21429-Alexa Fluor 555 Rb-IgG, ThermoFisher Scientific), counterstained with DAPI (blue). Pictures were obtained using a Zeiss or a Nikon eclipse Ti2 microscope, with magnification as indicated and processed using ImageJ Fiji version 1.50. Densitometric analysis was obtained as previously described [75].

### 4.12. Statistical Analysis

Bar graphics and statistical comparisons were obtained using GraphPad Prism 8.2.0. Comparative analyses between experimental groups were performed using unpaired Student’s *t* test and Mann-Whitney U test. Results were considered significant if the *p* value < 0.05. Data are expressed as mean ± SEM. Gene enrichment analyses from significantly altered genes were conducted using Enrichr v.2.0 software (http://amp.pharm.mssm.edu/Enrichr/, accessed on 2 November 2022) and https://reactome.org, accessed on 2 November 2022.

## Figures and Tables

**Figure 1 ijms-24-01453-f001:**
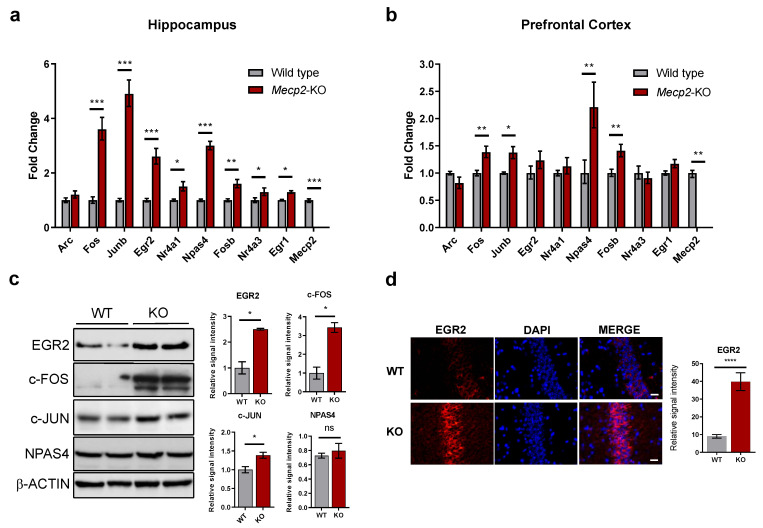
Validation of IEGs expression in *Mecp2*-null mice. (**a**,**b**) IEGs expression in the hippocampus and prefrontal cortex of 8-weeks mice. Graphs show mean ± SEM of *n* = 5–6 animals per condition. Student’s *t*-tests were used (* *p* < 0.05, ** *p*  <  0.01, *** *p* < 0.001). (**c**) Western Blot analysis of EGR2, c-FOS, c-Jun and NPAS4 in the hippocampus of 8-weeks mice. β-ACTIN was used as loading control. Samples from two animals per condition are shown. Graphs on the right represent quantitation of band intensity (mean values ± SEM, two tailed Mann-Whitney test, * *p* < 0.05, ns = not significant). (**d**) Representative immunostaining of EGR2 protein (red) in the hippocampal regions of WT or *Mecp2*-KO 8-week mice, counterstained with DAPI (blue). All images were processed with ImageJ Fiji version 1.50 g. Scale bar represents 200 µm. The graph corresponds to quantitation of *n* = 10 images per condition (mean values ± SEM, two-tailed unpaired *t* test, **** *p* < 0.001).

**Figure 2 ijms-24-01453-f002:**
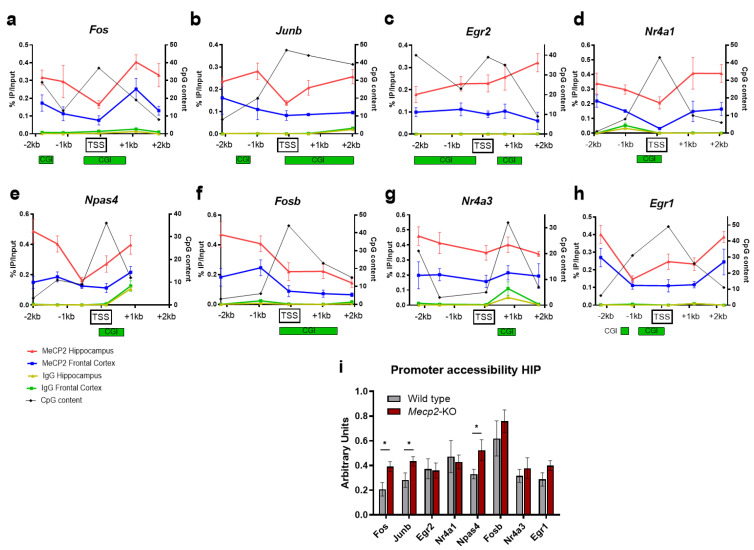
MeCP2 has a potential repressive role on IEGs. (**a**–**h**) Five regions encompassing each gene’s transcription start site were analyzed with quantitative chromatin immunoprecipitation–PCR (ChIP–qPCR) in the hippocampus of 8–week WT mice. Green boxes below each graph represent CpG islands (taken from UCSC genome browser, GRCm39/mm39). Values are expressed as the percentage of MeCP2 or IgG enrichment over the input on the left *y*–axis (*n* = 3–4 biological replicates for each condition, means ± SEM are represented), and the number of CpG dinucleotides in 500–bp windows on the right *y*–axis. (**i**) Micrococcal nuclease (MNase) accessibility assay in WT vs. *Mecp2*–KO hippocampi. MNase–digested DNA was subjected to quantitative PCR and normalized with input DNA (*n* = 3 biological replicates for each condition, means ± SEM are represented, * *p* < 0.05 in Student’s *t*-tests).

**Figure 3 ijms-24-01453-f003:**
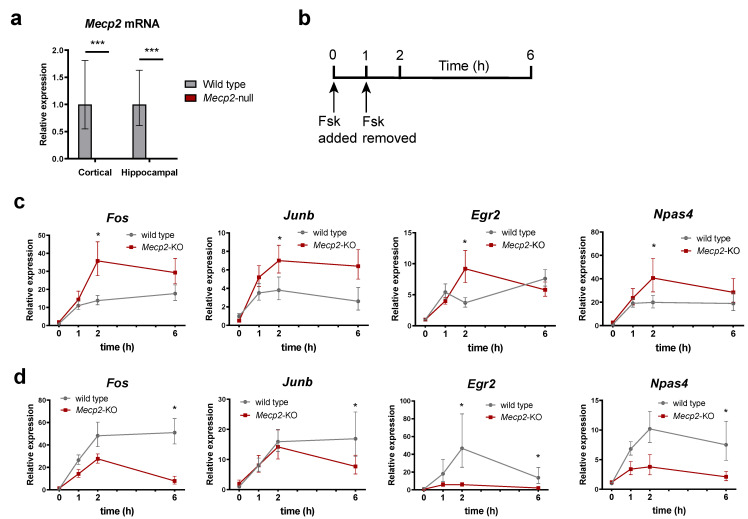
IEGs gene expression response to Forskolin is dysregulated in 8-weeks *Mecp2*-KO mice neurons. (**a**) *Mecp2* expression in cortical and hippocampal neurons, as measured by RT-qPCR. (**b**) Schematics indicating treatment of cultured primary neurons with forskolin and time-course of sample collection. (**c**,**d**) Time-course of *Fos*, *Junb*, *Egr2* and *Npas4* transcript levels analysis for both WT and *Mecp2*-KO hippocampal (**c**) or cortical (**d**) neurons upon treatment with forskolin. All expression data are relative to WT unstimulated values (*n* = 3 biological replicates for each condition, means ± SEM are represented, * *p* < 0.05 and *** *p* < 0.001 in Student’s *t*-tests).

**Figure 4 ijms-24-01453-f004:**
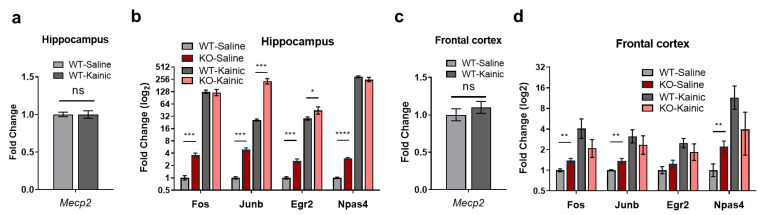
IEGs expression in WT and *Mecp2*-KO, 8-week mice following kainic acid (KA) administration. (**a**,**c**) *Mecp2* transcript levels in the hippocampus (**a**) or frontal cortex (**c**) of KA-treated versus untreated WT mice. (**b**,**d**) Expression levels of *Fos*, *Junb*, *Egr2* and *Npas4* one hour after KA administration in the hippocampus (**b**) or frontal cortex (**d**) of WT and *Mecp2*-KO mice (*n* = 5–6 animals for each condition, graphs represent means ± SEM, * *p* < 0.05, ** *p* < 0.01, *** *p* < 0.001 and **** *p* < 0.0001 in Student’s *t*-tests).

**Figure 5 ijms-24-01453-f005:**
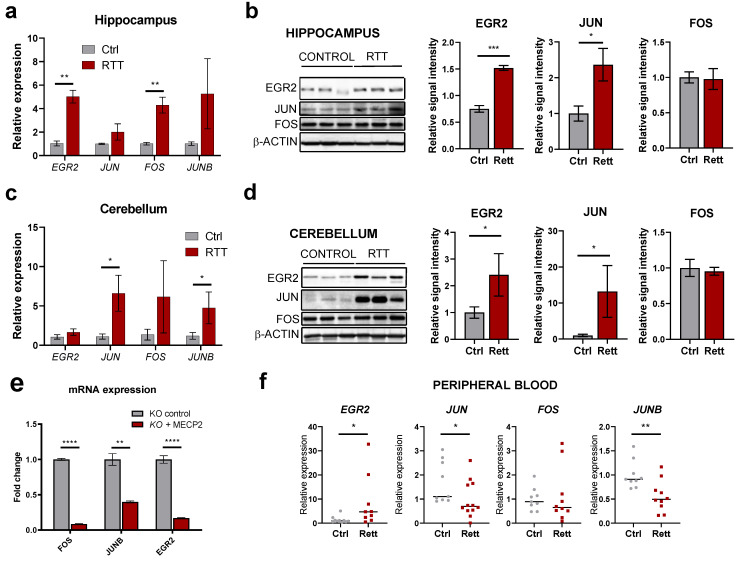
IEGs expression in human RTT patients and cellular models. (**a**,**c**) RT-qPCR analysis of *EGR2*, *JUN*, *FOS* and *JUNB* transcript levels in the hippocampus (**a**) or cerebellum (**c**) of post-mortem RTT patients or healthy control samples (*n* = 3 subjects per condition, graphs represent means ± SEM, * *p* < 0.05, ** *p* < 0.01 in Student’s *t*-tests). (**b**,**d**) Western blot analysis of EGR2, JUN and FOS protein levels in the hippocampus (**b**) or the cerebellum (**d**) of post-mortem RTT patients or healthy control samples. Graphs on the right indicate quantitation of band intensity (mean ± SEM, * *p* < 0.05, *** *p* < 0.001 in Student’s *t*-tests) of the three biological replicates. (**e**) RT-qPCR analysis of *FOS*, *JUNB* and *EGR2* expression upon overexpression of MeCP2 in a human neural progenitor model (*n* = 3 biological replicates, graphs represent means ± SEM, ** *p* < 0.01, **** *p* < 0.0001, in Student’s *t*-tests). (**f**) RT-qPCR analysis of *EGR2*, *JUN*, *FOS* and *JUNB* transcript levels in the peripheral blood of RTT patients or healthy control samples (*n* ≥ 10 biological replicates, graphs represent means ± SEM, * *p* < 0.05 and ** *p* < 0.01 in Student’s *t*-tests).

**Figure 6 ijms-24-01453-f006:**
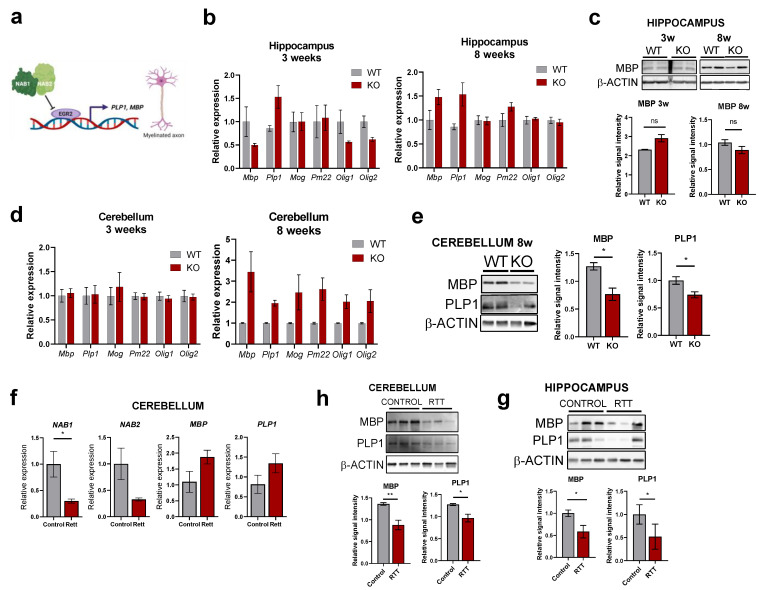
Expression levels of myelin-related genes in RTT mouse models and human patients. (**a**) The NAB1/2-EGR2 axis in the control of myelination. Dysregulation in either of these proteins results in abnormal myelination. Created with BioRender.com. (**b**,**d**) myelination markers transcript levels in the hippocampus (**b**) or cerebellum (**d**) of asymptomatic (3-weeks) or symptomatic (8-weeks) WT and *Mecp2*-KO animals were assessed by RT-qPCR. (*n* = 4 animals for each condition, graphs represent means ± SEM, * *p* < 0.05, ** *p* < 0.01, ns = not significant in Student’s *t*-tests). (**c**,**e**) Western Blot analysis of MBP and PLP1 proteins in the indicated regions of WT or *Mecp2*-KO mice (*n* = 2 animals). Graphs on the right indicate quantitation of band intensity (mean ± SEM, * *p* < 0.05, ns = not significant in Student’s *t*-tests). (**f**) *NAB1*, *NAB2*, *MBP* and *PLP1* mRNA levels in the cerebellum of post-mortem RTT patients or healthy control samples (*n* = 3 subjects per condition, graphs represent means ± SEM, * *p* < 0.05 in Student’s *t*-tests). (**h**,**g**) Western blot analysis of MBP and PLP1 protein levels in the cerebellum (**h**) or the hippocampus (**g**) of post-mortem RTT patients or healthy control samples (*n* = 3 subjects per condition). Graphs below indicate quantitation of band intensity (mean ± SEM, * *p* < 0.05, ** *p* < 0.01 in Student’s *t*-tests) of the three biological replicates.

**Table 1 ijms-24-01453-t001:** IEGs transcriptional changes in prefrontal cortex and hippocampus of *Mecp2*^y/−^ mice. n.d = not differentially expressed.

			Frontal Cortex		Hippocampus	
Gene Symbol	Gene Name	Transcript ID	Fold Change (KO vs. WT)	Adjusted *p*-Value	Fold Change (KO vs. WT)	Adjusted *p*-Value
*Arc*	Activity-regulated cytoskeleton-associated protein	uc007wfo.1	3.71	0	2.24	2.36 × 10^−83^
*Fos*	Proto-oncogene c-Fos	uc007oha.1	2.85	5.46 × 10^−36^	2.70	2.97 × 10^−18^
*Junb*	Transcription factor jun-B	uc012ghn.1	2.61	2.33 × 10^−17^	1.81	8.47 × 10^−10^
*Egr2*	Early growth response protein 2	uc007flx.1	2.39	5.03 × 10^−7^	4.29	2.76 × 10^−9^
*Npas4*	Neuronal PAS domain-containing protein 4	uc008gbu.2	2.30	4.51 × 10^−14^	3.60	2.64 × 10^−21^
*Nr4a1*	Nuclear receptor subfamily 4 group A member 1	uc007xsv.2	2.29	3.78 × 10^−63^	1.84	3.96 × 10^−25^
*Fosb*	Protein fosB	uc009flk.1	1.96	4.59 × 10^−8^	1.90	2.59 × 10^−5^
*Egr1*	Early growth response protein 1	uc008elt.1	1.70	6.69 × 10^−95^	1.60	8.24 × 10^−40^
*Nr4a3*	Nuclear receptor subfamily 4 group A member 3	uc008suw.1	1.39	4.29 × 10^−4^	n.d	n.d

## Data Availability

Raw data from RNA-seq experiments can be downloaded from https://www.ncbi.nlm.nih.gov/geo/query/acc.cgi?acc=GSE60219, accessed on 30 September 2018.

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
