# Peer review of "Global Impairment of Immediate-Early Genes Expression in Rett Syndrome Models and Patients Linked to Myelination Defects"

_ijms, 2023, doi:10.3390/ijms24021453_

Round 1

Reviewer 1 Report

Rett syndrome (RTT) is a severe neurodevelopmental disease caused almost exclusively by MeCP2 gene mutation. In the manuscript entitled “Global impairment of immediate-early genes expression in Rett syndrome models and patients linked to myelination defects”,  Sonia Guil et. al showed that MeCP2 directly bind several immediate-early genes at the chromatin level and upregulated in the hippocampus and prefrontal cortex of the Mecp2-KO mouse by using RNA-sequencing analysis. Furthermore, altered IEGs levels were found in RTT patient’s peripheral blood and brain regions of post-mortem samples. However, the mechanisms by which the transcription factor MeCP2 so profoundly affects activity-dependent expression programs and correct neural connections in the mammalian brain is yet to be determined. This study uncovers a transient mechanism of MeCP2 in the  proper regulation of IEGs expression and correct synaptic development. 

Major issues:

1) Fig1a,b. the Arc expression in hippocampus and cortex is not consistent with RNA-seq data, please explain. Fig1d, please add a neural cell marker co-staining with EGR2.

2) Fig2, an overexpression of Mecp2 in relevent cell will be necessary to show MeCP2 has a potential repressive role on IEGs.

3) Fig2, It will be helpful to show the Mecp2 binding on those genes from published Mecp2 chIP-seq data.

4) Fig5e, it is interesting to find that EGR2, JUN, FOS and JUNB transcript levels 

in the peripheral blood of RTT patients or healthy control samples. However, this is a nonneural tissue/cells, it is hard to link to the main topic in this paper.

5) Fig6, only Mbp and Plp gene expression is not enough to support the myelination defects claim. It is also hard to understand why elevated Mbp and Plp expression indicated myelination defects.

Minor issue:

Please provide a supplemental table on the gene expression level of those IEGs.

Reviewer 2 Report

Studies in this very interesting manuscript were aimed at better understanding the role of MeCP2, a transcription factor encoded by a gene mutated in patients affected by the severe neurodevelopmental disorder known as Rett syndrome. RNA-seq analysis, RT-qPCR and western blotting of hippocampus and prefrontal cortex of symptomatic 8-week-old Mecp2 knockout mice indicated dysregulation of multiple early immediate-early response genes (IEGs). Following chromatin immunoprecipitation and elegant chromatin accessibility assays strongly suggested that MeCP2 plays a direct repressive role in the expression of IEGs of importance for brain functioning. Furthermore, the authors found that IEGs dysregulated in the Rett mouse model were also affected in human samples from Rett patients.  Dysregulation of RGR2 expression and the NAB1/NAB2/EGR2 axis was in particular correlated with altered myelination. 

The extensive experiments are well designed and the manuscript is clearly written and well organized. There is no doubt that these studies are highly significant to the understanding of neural roles of MeCP2 and their implications in Rett syndrome.

However, there are some points that require further clarification, in particular those related to myelination:

1) The studies on myelination in the KO mouse did not detect significant alterations of MBP and PLP1 transcript levels in 21 day-old animals, a time of peak myelinating activity in the rodent brain. On the other hand, analysis at 8 weeks of age (when myelination normally reaches low basal levels) indicates dramatically elevated levels of myelinating gene activity. It would be important to correlate mRNA levels of MBP and PLP1 protein expression at both ages of development.  

2) The studies analyzing cerebellar MBP and PLP1 mRNA levels (Figure 6 d) need to indicate the exact age of the Rett and control individuals from which post-mortem samples were obtained. This is crucially important for correct comparison because myelination is a developmentally controlled process.

3) Discussion and inclusion of recent papers on MeCp2 /Rett and myelin appear to be missing.

- A previous publication that also found alterations in the expression of some of the IEGs reported in this paper appears to be missing (Osenberg.............Ballas. PNAS 115 (23), E5363–E5372, 2018).

Reviewer 3 Report

In the manuscript entitled “Global impairment of immediate-early genes expression in Rett 2 syndrome models and patients linked to myelination defects”, authors first identified a group of immediate-early genes (IEGs), which were up-regulated in the hippocampus and prefrontal cortex of MeCP2 hemizygous null male (MeCP2-/y, KO) mice. And then they demonstrated that MeCP2 protein was enriched in the TSS sites of those IEGs probably to repress their expression. In addition, the expression of Junb and Egr2 was abnormal in the mutant primary neuronal cultures treated with forskolin in vitro and in the mutant hippocampus treated with Kainic acid in vivo. Moreover, the expression of some IEGs was also mis-regulated in the hippocampus, cerebellum, and even peripheral blood of Rett syndrome patients. The compromised NAB1-2/EGR2 axis was a possible cause for the aberrant myelination in MeCP2 -KO mice. Overall, the findings are interesting. However, the following issues should be address.

Major comments:

1, Synaptic development and myelination are two different biological events. Authors claimed that proper IEGs expression is crucial for correct synaptic development; nonetheless, their evidence is not sufficient to support the point. Among several IEGs mentioned in the manuscript, there are Arc and Npas4, which are associated with post-synapse and inhibitory synapses, respectively. Unfortunately, the altered expression of Arc is not significant in the RNA-seq data of prefrontal cortex; moreover, the expression of Arc is not altered in both the hippocampus and prefrontal cortex by quantitative PCR assays. The expression levels of Npas4 is upregulated at both transcriptional and translational levels in MeCP2 mutant mouse. It is necessary to confirm the increased expression of Npas4 in the hippocampus by immunofluorescence assays with mouse samples in Fig1, and by PCR or western blot assays with patient samples in Fig.5. In addition, it would be helpful, if authors could add another IEG gene associated with synaptic development, as Npas4.

2, Authors listed several IEGs including Arc, Fos, Junb, Egr2, Npas4, Nr4a1, Fosb, Egr1, and Nr4a3 in Table 1. However, they failed to provide clear approach to reveal how they identified those IEGs from RNA-seq data. It is essential to show detailed processes how those IEGs is identified in systematic RNA-seq assays unbiasedly.

Minor comments:

1, In Fig.1d, authors performed immunofluorescent staining to show the expression of EGR2 in the hippocampus in both the control and MeCP2 mutant. Hippocampus is a heterozygous structure with CA and dentate gyrus (DG) domains; however, it is not clear whether EGR2 is upregulated in CA or/and DG domain in the mutant. Authors should show the expression profiles of EGR2 in the hippocampus with both low and high magnificence?

2, In Figs. 1d, 2, and 4, the age of the experimental mice used should be specified.

3, In Fig. 5b, the western blot image of EGR2 is not consistent with the quantitative data.

4, In Fig. S2, for the consistency, author should add transcriptional data of gene expression in the frontal cortex?

Round 2

Reviewer 3 Report

Good job. Almost all the issues raised were addressed appropriately.

Author Response

We thank the reviewer for the positive appreciation of our revised manuscript.

As suggested, the manuscript has now been checked by a native English-speaking scientist and all changes introduced in the revised file appear in blue.